# Sesame as an Alternative Host Plant to Establish and Retain Predatory Mirids in Open-Field Tomatoes

**DOI:** 10.3390/plants11202779

**Published:** 2022-10-20

**Authors:** Jose Castillo, Amy Roda, Jawwad Qureshi, Meritxell Pérez-Hedo, Alberto Urbaneja, Philip Stansly

**Affiliations:** 1Department of Entomology and Nematology, Southwest Florida Research and Education Center, University of Florida, Immokalee, FL 34142, USA; 2United States Department of Agriculture, Animal and Plant Health Inspection Service, Plant Protection and Quarantine, Science and Technology, Miami, FL 33158, USA; 3Instituto Valenciano de Investigaciones Agrarias (IVIA), Centro de Protección Vegetal y Biotecnología, CV-315, Km. 10, 7, 46113 Moncada, Valencia, Spain

**Keywords:** preventive biological control, companion plant, invasive pests, *Bemisia tabaci*, *Tuta absoluta*, *Nesidiocoris tenuis*, *Macrolophus praeclarus*

## Abstract

The silverleaf whitefly (*Bemisia tabaci*) and the South America tomato pinworm (*Tuta absoluta*) are two of the most destructive pests of tomato. Open-field tomato production frequently relies on chemical treatments, which has been shown to lead to pesticide resistance. The integration of biological control using predatory mirid bugs is an effective alternative method for managing these pests. However, methods to establish and maintain populations of zoophytophagous mirids are not adequately described. We explored the potential use of two mirids naturally occurring in Florida, *Nesidiocoris tenuis* and *Macrolophus praeclarus*. We conducted 6 field experiments over 4 consecutive years to develop a strategy to maintain the mirids. Pre-plant inoculation of tomato plants did not lead to their establishment, likely due to the low prevalence of prey. We explored the use of sesame (*Sesamum indicum*) to retain the mirids. Intercropping sesame maintained the populations of *N. tenuis* throughout the duration of the crop. *Macrolophus praeclarus* never established in any of the open-field experiments. *Nesidiocoris tenuis* damage was minimal (<1 necrotic ring/plant) and mirid damage was reduced in the presence of sesame. Our results show that intercropping sesame may provide a means to utilize mirids to manage *B. tabaci,* an established pest, and provide options to tomato growers should *T. absoluta* invade USA.

## 1. Introduction

Tomato, a crop of major economic importance [1], is under constant threat from arthropod pests that can cause severe losses. Among these, the siverleaf whitefly, *Bemisia tabaci* (Gennadius) (Hemiptera: Aleyrodidae) and the South America tomato pinworm, *Tuta absoluta* (Meyrick) (Lepidoptera: Gelechiidae) are currently two of the most destructive and difficult pests to manage [2,3]. *Bemisia tabaci* has invaded the United States, Australia, Africa, and several European countries. *Bemisia tabaci* causes damage through feeding and, more importantly, as a vector of viruses [3]. *Tuta absoluta*, a pest native to South America, invaded the Mediterranean region in 2006 causing high yield losses [4]. From there, *T*. *absoluta* quickly spread across the Afro-Eurasian supercontinent, Central America, and China, and threatens to invade Mexico and the United States, which together account for 25% of the world’s tomato crop area and 42% of the world’s tomato production [2].

Insecticide application continues to be the primary control method for these pests in most parts of the world [2]. Chemical tools can provide relief, but alternatives are needed because of insecticide resistance and sub-lethal effects [5,6], the ever-increasing close interface with suburban communities, environmental contamination, worker and food safety issues, and prohibitive costs [7,8,9]. The fact that management of these pests solely based on chemical control has led to insecticide resistance and control failure [10,11] has encouraged the development of non-chemical alternatives (i.e., botanical insecticides, trapping, use of pheromones, cultural control, and biological control) [2]. 

The use of predatory mirids has been found to be one of the most successful alternatives to chemical insecticides [12]. *Nesidiocoris tenuis* Reuter and *Macrolophus pygmaeus* Rambur (Hemiptera: Miridae) are widely used for biological control of *B. tabaci* and *T. absoluta* in European greenhouses [13,14]. Additionally, the plant feeding of these zoophytophagus mirids can activate defensive responses in the tomato plant, which makes the plant less susceptible to the attack [15,16,17,18].

However, mirids can only regulate pest populations if they are well established in the crop [12]. There are two release strategies based on the level of pest pressure at the time of transplanting time. If pest pressure is high, mirids are released and allowed to establish on the tomato plants before transplanting [19,20]. Inoculation before transplanting tomatoes is common in southeastern Europe in summer planting cycles where the populations of *T. absoluta* and the whitefly *Bemisia tabaci* Gennadius (Hemiptera: Aleyrodidae) are high at the time of transplantation [21]. When pest pressure is low or non-existent, such as with crops with planting dates at the end of winter, mirids are released in the field 3–4 weeks after transplantation [22,23]. In this situation, the mirid populations increase as pests appear [21]. In both release strategies, the addition of alternative food, mainly the mixture of frozen *Ephestia kuehniella* Zeller (Lepidoptera: Pyralidae) eggs and *Artemia* spp. (Anostraca: Artemiidae) cysts is highly recommended to successfully establish the mirids. These strategies were designed for greenhouse tomato crops, where Integrated Pest Management (IPM) strategies based on the use of natural enemies have been widely developed to help reduce the excessive use of insecticides [24]. However, strategies for using mirids in open-field tomato production require further development.

Multiple factors limit the use of biological control agents in open-field tomato production. One major drawback is that the natural enemies will likely leave the crop at low pest levels. The phytophagous behavior of predatory mirids may overcome this barrier by helping to retain the predator in the absence of prey. However, heavy plant feeding may lead to economic damage [25]. When confined in greenhouses, *N. tenuis* can become a pest of tomatoes when prey is scarce by causing necrotic rings that can lead to plant stunting, flower abortion, and, ultimately, reduced yield [26]. Management strategies based on selective pesticide applications, the addition of supplemental food, and companion planting with sesame [*Sesamum indicum* L. (Lamiales; Pedaliaceae)] have been developed to lessen the mirids’ negative impact and to enhance their establishment in protected tomatoes [27,28,29,30,31]. Another approach has been to use other mirid species, such as *Macrolophus* spp. or *Dicyphus* spp. that are not considered pests of tomatoes [17,32,33,34]. To date, the damage caused by *N. tenuis* in commercial open-field tomato production remains unknown. 

Cultural control measures have been developed for open-field systems, which include use of resistant varieties and repellents, crop rotation, the removal and destruction of infested plant material and the use of reflective mulch [35,36]. Although there have been efforts to use parasitoids to control whiteflies [37], to date the use of natural enemies in open-field tomato has been neglected. 

The aim of this study was to develop methods to deploy zoophytophagus mirid in open-field tomatoes to control established pests such as *B. tabaci* and prepare for the potential invasion of *T. absoluta*. The studies were conducted in Florida where tomatoes are produced mainly in large open-field areas [38]. The main pest that attacks Florida tomatoes is the whitefly *B. tabaci* [39]. An invasion of *T. absoluta* is anticipated to cause major economic damage based on the losses that were experienced in Europe, Israel and Africa and the difficulty in controlling the pest with insecticides. We tested two established mirids in southern USA, *N. tenuis* and *Macrolophus praeclarus* (Distant), that were previously tested in laboratory and field cage studies to be potential biological control agents of *B. tabaci* and pest moths [17,32]. The studies showed that *M. praeclarus* did not cause feeding damage to tomato plants compared to *N. tenuis* and that adding sesame increased *N. tenuis* populations without increasing damage to the tomatoes. Although the addition of sesame also slightly increased *M. praeclarus* numbers in the absence of prey, this species did not reach the adult stage by feeding on sesame alone. In this study, we explored the use of sesame to retain and increase populations of unconfined mirids. Our main goal was to examine the mirids’ effectiveness in controlling populations of an established pest, *B. tabaci,* and to learn how to exploit them for developing an IPM program for the possible incursion of *T. absoluta*. 

## 2. Results 

### 2.1. Pre-Plant Inoculation of Tomato to Establish N. tenuis in Open Field Tomatoes 

*Nesidiocoris tenuis* dispersed early from the field plots, which corresponded to the decrease in the availability of *B. tabaci* as prey (Figure 1C,D). The number of *B. tabaci* found in plots receiving *N. tenuis* was similar to both the insecticide-treated and the untreated plots (Figure 1B) and there was no interaction between week and treatment (F_14,63_ = 1.17 *p* = 0.323). Mirid feeding reduced the subsequent generations of *B. tabaci* as seen with lower numbers of eggs and nymphs 7 and 8 weeks after release, but their control did not persist (Figure 1B). Significantly fewer *B. tabaci* eggs and nymphs were seen on plants treated with insecticides compared to the untreated plots (Figure 1B, treatment: F_2,9_ = 5.85 *p* < 0.024) as the populations of *B. tabaci* increased during the study (week: F_7,63_ = 12.53 *p* < 0.001). The mirids did little damage to the tomatoes in accordance with their low numbers. Necrotic rings were observed only on one date (16 March 2015) and on only three plants out of the total 140 plants sampled.

In contrast, *N. tenuis* was more effective and more damaging when confined to cages (Appendix A). The mirids rapidly decimated the number of *B. tabaci* (treatment: F_2,9_ = 53.04 *p* < 0.001), which remained near zero for the duration of the study (week: F_4,36_ = 1.79 *p* = 0.152, treatment x week F_8,36_ = 7.23 *p* < 0.001). The mirids controlled *B. tabaci* numbers to similar levels as the insecticide treatment, which was significantly lower compared to cages with *B. tabaci*-infested tomatoes (S1B, treatment: F_7,24_ = 3.13 *p* = 0.017). Unlike the field study, the number of mirids was consistent throughout the experiment (S1C, F_7,24_ = 3.13 *p* = 0.017). When dispersal was not an option, the number of necrotic rings on the tomatoes remained similar through the course of the experiment (F_3,12_ = 0.41 *p* = 0.748). 

### 2.2. Tomato-Sesame Intercrop to Retain Zoophyatophagus Mirids

#### 2.2.1. Field Evaluation of Sesame to Retain *N. tenuis* in Tomato

The addition of sesame to a tomato field planting retained *N. tenuis* for the duration of the study even when whitefly numbers were very low (Figure 2B). The mirids were able to control the initial population of whiteflies as well as natural migration into the field. The mirids reduced the number of whiteflies to near zero levels and the number of whiteflies did not increase during the study (Figure 2B, F_3,12_ = 0.24 *p* = 0.865). Similar numbers of whiteflies were found on sesame and tomato plants and there were no differences whether the tomato was planted next to sesame or planted next to tomato. Mirids were found on both tomato and sesame plants throughout the study. Mirids were found more frequently in the tomato release plots planted next to tomatoes and in sesame release plots planted next to tomatoes compared to sesame plants planted next to sesame plants and tomato planted next to sesame (Figure 2C, F_3,12_ = 11.36 *p* < 0.001). The effect of interplanting sesame on tomato damage changed during the study (Figure 2D, F_9,108_ = 18.52 *p* < 0.001) and there was no interaction between week and treatment (F_9,108_ = 1.56 *p* < 0.06). Initially, more tomato seedlings in the sesame-tomato intercrop had necrotic rings compared to tomato seedlings planted next to tomato. However, the early damage to tomato in the tomato-sesame intercrop was not detectable as the plants grew. In fact, more damaged tomato plants were seen when tomato was planted next to tomato when populations of mirids were high in November compared to the sesame-tomato intercrop that had similarly high numbers of plants with mirids (Figure 2C), showing the potential benefit of sesame to lower damage to tomatoes as mirid populations increase (Figure 2D). 

#### 2.2.2. Field Evaluation of Reduced Amount of Sesame in Tomato-Sesame Interplanting 

The spring 2016 field study showed that the amount of sesame needed to retain mirids in the crop could be greatly reduced. With a border of 10 sesame plants, the mirids were retained in the tomato plants even when *B. tabaci* prey were reduced to near zero (Figure 3). In fact, the sesame -tomato planting was sufficient to retain the mirids in the tomato-alone planting. There was no *N. tenuis* damage seen on tomato plants. There were no differences in the number of *B. tabaci* found in the tomato alone planting and sesame-tomato planting (Figure 3B, F_1,8_ = 4.47 *p* = 0.066) and there was an interaction between week and treatment (F_6,65_ = 12.82 *p* < 0.001). More whiteflies were found on sesame on the first collection date compared to the tomato plants in the tomato alone or the tomato-sesame intercrop. However, the number of whiteflies was reduced to similar levels as seen on tomato planted alone or in the tomato-sesame intercrop (Figure 3B). Mirids were seen on sesame and tomato plants 5 weeks after the plants were transplanted into the field and remained at similar levels throughout the remainder of the study (Figure 3C. week: F_6,54_ = 1.76 *p* = 0.117). There was no difference in the percentage of tomato plants with mirids in the tomato-sesame intercrop or the tomato alone planting and there was no difference in the percentage of sesame and tomato plants with mirids (Figure 3C, treatment: F_2,9_ = 0.533 *p* = 0.604) and there was no interaction between week and treatment (F_12,54_ = 0.413 *p* = 0.952).

#### 2.2.3. Field Comparison of Pesticide and *N. tenuis* in a Tomato-Sesame Intercrop to Control *B. tabaci*


The fall 2016 field study showed that *N. tenuis* released into a tomato-sesame intercrop plots controlled *B. tabaci* populations on tomato to similar levels found in pesticide-treated plots (Figure 4B). The experiment also showed the highly mobile nature of the mirids. *Nesidiocoris tenuis* were found in the untreated tomato plots and insecticide-treated sesame 6 weeks after planting (Figure 4C,E). This invasion of *N. tenuis* into the other treatments corresponded to an increase in *B. tabaci* prey and a subsequent decrease in the number of *B. tabaci* in untreated plots. *Bemisia tabaci* numbers increased for 9 weeks and then decreased in the subsequent 4 weeks of the study (Figure 4B, week: F_12,108_ = 25.39 *p* < 0.001, week x treatment: F_24,108_ = 1.43 *p* = 0.109). There were no significant differences in the number of *B. tabaci* found on tomatoes in untreated, insecticide-treated and *N. tenuis* release plots (F_2,9_ = 2.40 *p* = 0.146). The percentage of tomato plants with mirids increased during the study and there were no differences between the amount of tomato plants with mirids in untreated and mirid release plots (Figure 4C). 

*Bemisia tabaci* and *N. tenuis* were found on both the untreated and insecticide-treated sesame (Figure 4D,E). Like the tomato plots, *B. tabaci* numbers were low throughout the experiment and there were no differences in the number of *B. tabaci* found in the insecticide-treated and untreated plots (F_1,6_ = 3.75, *p* = 0.101). *Bemisia tabaci* numbers fluctuated during the study with a significant effect of week (F_12,72_ = 5.64, *p* =< 0.001) seen, and the lowest number of *B. tabaci* were found during the last 3 weeks of the study (Figure 4D). Although the insecticide application did not completely exclude the presence of *N. tenuis* on sesame, mirid populations were reduced on insecticide-treated sesame with their presence being recorded on only one sample date (28 November 2016).

#### 2.2.4. Field Evaluation of *M. praeclarus* to Control *B. tabaci* in a Tomato-Sesame Intercrop 

*Macrolophus praeclarus* were never recovered from tomato alone or the tomato-sesame intercrop. *Bemisia tabaci* numbers increased in the absence of the mirids. *Bemisia tabaci* numbers were low for the first 5 sample dates then increased in the subsequent 3 sample dates (week: F_7,84_ = 183.8 *p* < 0.001; week x treatment: F_21,84_ = 1.45 *p* = 0.119).

Although not released, *N. tenuis* moved into the experimental plots from the surrounding area on the last sampling day. *Nesidiocoris tenuis* were found in all treatments and there was no effect of planting (tomato alone or tomato-sesame intercrop; F_1,12_ = 0.016, *p* = 0.901) on the percentage of tomato plants with mirids found in the plots (Figure 5C). 

#### 2.2.5. Field Comparison of Pesticide, *M. praeclarus* and *N. tenuis* to Control *B. tabaci* in Tomato Alone and Tomato-Sesame Intercrop

Mirids reduced whiteflies on tomatoes as there were fewer nymphs and eggs in plots with mirids compared to the untreated plots (Figure 6B, F_4,21_ = 40.31 *p* < 0.001). The difference was most evident during the first 5 weeks of the experiment, where the whitefly numbers were reduced by 40–75 % in treatments with mirids compared to untreated tomatoes (week: F_7,139_ = 41.67 *p* < 0.001, week x treatment: F_28,139_ = 2.39 *p* = 0.001). Plots treated with insecticides had the lowest number of whiteflies, which was consistent through the study (Figure 6B). 

The highly mobile nature of mirids was very apparent in this experiment. During the first 5 weeks of the study, mirids were present only in plots where mirid releases were made (Figure 6C). By week 6, *N. tenuis* was found in all the untreated plots and the *M. praeclarus* release plots and by week 8 they were present in the pesticide-treated plots. *Macrolophus praeclarus* were not recovered from the *M. praeclarus* release plots or from any of the other treatment plots. The establishment of *N. tenuis* in the untreated treatments paralleled a decrease in *Bemisia tabaci* numbers (Figure 6). 

Mirid damage was found in all treatments (Figure 6D). However, the damage was low, with a mean of less than one ring being seen per plant. Corresponding to mirid abundance (Figure 6C), the number of rings found on tomatoes was highest in treatments with mirid releases (F_4,20_ = 6.97, *p* = 0.001). The presence of sesame reduced *N. tenuis* feeding damage (Figure 6D). Pesticides reduced but did not totally prevent *N. tenuis* damage. *Nesidiocoris tenuis* and necrotic rings were found on pesticide-treated tomato plants on the last two sample dates.

## 3. Discussion

The use of zoophytophagous predators in commercial vegetable production has increased worldwide [24]. However, their use in open tomato field cropping systems is very limited compared to greenhouse production. Our results showed that zoophytophagus mirids can be established in open field crops and they can provide control of *B. tabaci* when other management strategies were also in place. These results lead to the possibility of developing an IPM strategy utilizing mirids as biological control agents in open-field tomatoes. 

Inoculating tomato seedlings with mirids prior to field planting was critical to establish the predator in the field. In our initial field studies, we never recovered *N. tenuis* when adults were released directly into plots. With low prey densities, the mirids quickly dispersed and providing supplementary food to establish the mirids on tomato seedlings prior to planting was found to be necessary. 

Utilizing mirids’ phytophagous behavior appears to be a viable means to retain them when prey numbers are low. With the addition of sesame as a companion crop, *N. tenuis* was present the entire cropping season even when *B. tabaci* populations were near zero. *Nesidiocoris tenuis* can reproduce on sesame in the absence of prey [40]. In cage studies, the combination of sesame and tomato increased *N. tenuis* populations without increasing damage to the tomatoes [17,28,31]. This study showed that the benefits previously seen with *T. absoluta* when mirids, *N. tenuis* and *Macrolophus pygmaeus*, were confined to a cage [31] could also work to control *B. tabaci* in open-field production where the mirids could leave the crop. When given the ability to move freely in the field, *N. tenuis* did not remain on sesame despite the presence of whiteflies and being a better plant resource than tomato [27]. The mirid was found equally on tomato and sesame plants and readily moved into to treatment plots containing only tomato. Mirids migrating into tomato crops have been found to concentrate on plants with high prey density, which may enhance pest control on tomato. In all four years, *N. tenuis* reduced the number of *B. tabaci* to very low numbers (<10 whiteflies/leaf), which were similar to numbers seen in the pots where pesticides commonly used by USA growers were applied. In fact, *N. tenuis* began to invade all treatments including the pesticide-treated pots once whitefly numbers began to increase. This indicates that *N. tenuis* may work effectively to control whiteflies in open fields. However, further work is needed to evaluate whether the mirids can provide control similar to pesticides in large fields, typical of USA, Mexico and China production. Our reduced sesame planting study showed that a number of sesame plants needed to retain the mirid could be decreased and still retain mirids in the plots. Additional studies are needed to determine the amount of sesame and planting distribution to have the mirids remain and effectively control *B. tabaci* in large commercial tomato fields. 

Over the course of this study, N. tenuis was found to establish near the experimental plots. In fact, the mirid invaded the M. praeclarus experiment despite not being released that season. Where N. tenuis is established, the mirid can fortuitously colonize crops [41,42,43]. This is advantageous in Southern Europe as *N. tenuis* controls *T. absoluta* [14]. However, their phytophagy can cause significant damage to tomatoes grown in heated greenhouses when pest populations are low [25,44]. Consequently, *N. tenuis* is considered a pest of heated greenhouse tomatoes in Northern Europe [45]. *Nesidiocoris tenuis* can also cause damage in open-field sesame and tobacco crops and is considered a pest [46,47]. Our cage studies and many other greenhouse studies showed that confining *N. tenuis* forces the mirid to rely on tomato plants for survival. Logically, the likelihood for damage increases. Our studies suggest that *N. tenuis* will disperse and cause minimal damage in open-field tomato cropping systems. In fact, we recorded very low levels of damage and had difficulty maintaining a population in the crop without the presence of sesame. Hence, the status of *N. tenuis* as pest or biological control agent will depend on the crop, pest complex, and cropping system.

Other species of mirids that cause less damage to tomato have been used to control pests in greenhouse production systems [13]. For the USA, *M. praeclarus* appeared to be a good biological control candidate. In laboratory and field cage studies, the mirid controlled *B. tabaci*, preyed on moth eggs (*Ephestia kuehniella*), and caused no damage when confined to tomato plants [17,32]. In addition, *M. praeclarus* is reported from South Florida, the Caribbean, Mexico, and Central America, making this species a potentially accessible biological control agent where *T. absoluta* has not invaded. However, different methods are needed to utilize *M. praeclarus* in open-field tomatoes. In our studies, *M. praeclarus* did not establish in tomato alone or in a sesame-tomato intercrop. On the contrary, *N. tenuis* invaded the *M. praeclarus* release plots in both experiments where *M. praeclarus* was released. Further studies are needed to explore whether *M. praeclarus* was not recovered because *N. tenuis* fed upon or behaviorally displaced *M. praeclarus* or whether other factors in the experiment (i.e., low density of prey or environmental conditions) caused the mirid to disperse. 

Pesticide resistance and environmental problems necessitate looking for alternatives to control important tomato pests like *B. tabaci* and *T. absoluta*. For successful biological control, predators must establish when the pest population is low [12,21]. As seen in our studies, *B. tabaci* invasions are unpredictable. The mirids’ generalist predation behavior has the benefit of controlling other pests, like tomato leafminers, should they invade the crop. Although *N. tenuis* and *Macrolophus* species are generalists, they show prey preference [47,48,49,50]. In fact, the need to provide *N. tenuis* supplementary food in order to retain the mirids to control *T. absoluta* is reduced when *B tabaci* are present [51]. Understanding the mirids’ prey preference is essential to select species for release that will likely attack the target pest(s) when given a choice. In addition, the mirid’s phytophagous feeding behavior can help to maintain the predator in the field in the absence of the target pest. We showed that a low density of sesame plants served as a method to retain the predator when prey densities were low. Additionally, non-cultivated host plants may provide refuges from pesticide residues [52,53]. Our studies were the first steps in learning how to incorporate zoophytophagus mirids to manage an existing pest, whiteflies, in open field tomatoes as well as prepare for potential new invaders such as *T. absoluta* that are spreading in the region.

## 4. Materials and Methods

### 4.1. Plants and Colonies 

*Nesidiocoris tenuis* and *M. praeclarus* were collected from *Uncarina grandidieri* (Baill.) Stapf (Lamiales: Pedaliaceae) in Miami, Florida, USA, and used to establish colonies on pesticide-free seedlings at the University of Florida, Southwest Florida Research and Education Center (SWFREC), Immokalee, FL. Tomato (var. ‘Lanai’ Tomato Growers Supply Company, Ft. Myers, FL, USA) was used to rear *M. praeclarus* and tobacco was used to rear *N. tenuis* as tobacco was more tolerant to feeding damage caused by this specie than tomato. The plants were seeded 4–5 weeks before use in the colonies or the experiments. Tomato plants (30 cm high) and tobacco plants (40–50 cm high) were placed into insect rearing cages (61 × 61 × 61 cm, Bugdorm MegaView Science Company, Ltd., Taichung Taiwan) and the mirids were subsequently added. Both mirid species were fed a mixture of frozen *E. kuehniella* eggs and *Artemia* spp. cysts (1:5 *w:w*) (Koppert Biological System, Howell, MI, USA), which was offered ad libitum on the adhesive portion of repositionable notes (Post-it^®^ Brand, 3M Cynthiana, KY, USA). Every 3 months, the tomato plants were cut at the base and placed on new plants to allow nymphs to move to the new material. Tobacco plants were trimmed every 30 days to cage height. *Bemisia tabaci* were collected from an abandoned tomato field at SWFREC and reared on collards (*Brassica oleracea* L. var. acephala). Colonies and plants were maintained at 25 ± 1 °C, 60% humidity and 14:10 h (L:D) photoperiodic conditions in climatic chambers at SWFREC.

Tomato and sesame (Hancock Seed Company, Dade City, FL, USA) used in the field experiments were seeded individually into 128 cell plug seed starting trays (8 × 16 cells each). Seedlings were fertilized once with Milogranite^®^ nitrogen fertilizer (0.5 g per cell, Milogranite^®^, Milwaukee, WI, USA) and 3 times weekly with Peters Professional 20-20-20 general purpose fertilizer (5 mL per 1 L of water). Tomato plants were used 4 weeks after seeding and sesame plants 18 days after sowing. 

### 4.2. Field Plot Preparation

Six field experiments were conducted to determine whether *M. praclearus* and *N. tenuis* could be established in open field tomato production systems and whether the mirids damaged the tomatoes. All trials were conducted at the IFAS /SWFREC research facility near Immokalee, Florida, on single row raised beds 81 cm × 128 m long and 20 cm high. The experiments were run during the south Florida tomato growing season; in the fall (October–January) and in the spring (March–May). Tomato varieties were selected to best match the planting season [54]. Granular 10-2-10 NPK fertilizer was incorporated before each planting at a rate of 121 kg N/ha. The remaining fertilizer requirements were met through fertigation over the course of the crop with liquid 7-2-7 NPK. Beds were then fumigated with 121 kg/ha 50:50 methyl bromide + chloropicrin, two drip tapes with 20.3 cm emitter spacing were laid down and beds were covered with whiteface (fall) or black (spring) polyethylene film mulch (fumigant barrier film, Raven VaporSafe^®^ 1.1 mil, Raven Engineered Film, Sioux Falls, SD, USA). The experimental rows of plants were separated by two rows (81 cm × 128 m) of corn (*Zea mays* L.). Experimental plots were arranged in a randomized complete block design (RCB). 

### 4.3. Comparison of N. tenuis and Pesticides to Control Whiteflies

#### 4.3.1. Pre-Plant Inoculation of Tomato to Establish *N. tenuis* in Open-Field Tomatoes

We conducted studies to determine whether *N. tenuis* would establish in open-field tomatoes by inoculating seedling tomatoes with the mirid and prey (*B. tabaci*). Tomato plants were inoculated with whiteflies and *N. tenuis* by placing tomato seedlings (TYLCV tolerant variety ‘BHN 8845′) into cages. To inoculate the tomatoes with *B. tabaci*, infested sesame plants (5–6) taken from the colony were placed in the cage to allow an estimated 10–15 adults per seedling to move to the tomato plants. For mirid treatment, 50 female and 50 male *N. tenuis* were randomly collected from the colony and introduced to a second cage with 100 *B. tabaci*-infested tomatoes as previously described. The mirids were provisioned with *E.*
*kuehniella* eggs/*Artemia* cysts and allowed to oviposit for 1 week. The seedlings were transplanted on 2 March 2015, into two rows of raised beds (46 cm spacing/35 plants per plot, Figure 1A), with each tomato plot separated by a plot of corn (10 plants/plot). Seedlings in the insecticide treatment plots received 1 application of imidacloprid (Admire^®^ Pro 207 mL/4047 m^2^, Bayer Crop Science LP, Research Triangle Park, NC, USA) applied in transplanting/setting water 7 days after transplanting. Whiteflies were also allowed to infest the experimental plots naturally. The experiment consisted of three treatments (n = 4): (1) *N. tenuis* + *B. tabaci*, (2) insecticide + *B. tabaci* and (3) untreated tomatoes + *B. tabaci*. Whitefly populations were monitored weekly by randomly selecting 5 plants from the middle of each experimental plot. One leaf was randomly selected from the top 25-50 cm of the plant. In the laboratory, all whitefly eggs and nymphs were counted using a dissecting microscope (10×). Mirids were monitored weekly by counting all mirid adults and nymphs in situ seen on each tomato plant in the *N. tenuis* + *B. tabaci* plots. 

#### 4.3.2. Cage Evaluation of Pesticide and *N. tenuis* to Control *B. tabaci*

To understand the impact of the mirids under confined conditions, the comparison of *N. tenuis* to pesticides experiment was repeated in cages (61 cm × 61 cm × 91 cm, Bugdorm MegaView Science Company, Ltd., Taichung Taiwan) held in a polycarbonate greenhouse provided with evaporative cooling (25 ± 1 °C, 60 % humidity and 14:10 h (L:D) photoperiodic conditions). This experiment used the same pool of whitefly and whitefly+ mirid-infested seedlings as were used for the field experiments. Four potted plants, taken from the same pool of plants used in the field study described above, were randomly assigned to cages, which were then arranged in a randomly completed block design (n = 4) in the greenhouse. As above, the experiment consisted of three treatments: (1) *N. tenuis* + *B. tabaci*, (2) insecticide + *B. tabaci* and (3) untreated tomatoes + *B. tabaci*. *Bemisia tabaci* populations were monitored weekly by taking 1 leaf from the middle of each plant. The leaves were taken to the laboratory and examined under the dissecting microscope (10×) and all whitefly eggs and nymphs were counted. *N. tenuis* populations and associated damage were monitored by randomly selecting one of the plants in the cage and counting all mirids (adults and nymphs) and necrotic rings in situ. 

### 4.4. Tomato-Sesame Intercrop to Retain Zoophyatophagus Mirids 

#### 4.4.1. Field Evaluation of Sesame-Tomato Interplanting to Retain *N. tenuis*

In the fall of 2015, a field experiment was conducted to determine whether sesame plants would help retain *N. tenuis* when planted near tomatoes. Tomato (var: BHN 8846) and sesame seedlings were pre-infested with whiteflies and mirids prior to transplanting in the fields. To infest seedlings with whiteflies, plants were confined in cages with highly infested sesame plants for 1 week. To inoculate the plants with mirids, 100 whitefly-infested tomato and sesame seedlings were placed in cages with 50 female and 50 male *N. tenuis* (1 mirid/plant) for 1 week. Additionally, the mirids were provided with *E. kuehniella* eggs and *Artemia* spp. cysts ad libitum. Each whole plot contained a total of 36 plants spaced 30.5 cm apart (Figure 2A). The center 12 plants were infested with *N. tenuis* (tomato or sesame) and whiteflies. On either side of the center planting were 12 whitefly-infested sesame or tomato (no *N. tenuis*) seedlings. There were 4 treatments arranged in a RCB (n = 4): (1) tomato, tomato + *N. tenuis*, tomato; (2) sesame, sesame + *N. tenuis*, sesame; (3) tomato, sesame + *N. tenuis*, tomato; (4) sesame, tomato + *N. tenuis*, sesame. Beginning 1 month after transplanting, the number of whitefly nymphs and eggs was evaluated each week by selecting 1 leaf from the mid-canopy of 3 randomly selected plants in the central planting. The presence and absence of *N. tenuis* were evaluated by counting the number of plants in the center sub-plot with at least 1 mirid per plant. *N. tenuis* damage was assessed by counting the number of tomato plants from the center planting (treatments 1 and 3) with visible necrotic rings. 

#### 4.4.2. Field Evaluation of Reduced Amount of Sesame in Tomato-Sesame Interplanting

In the spring of 2016, the amount of sesame interplanted with tomato was reduced to determine whether the lower amount would still help retain *N. tenuis*. Tomato (var: BHN 8845) and sesame seedlings were infested with *B. tabaci* by placing trays in cages with highly *B. tabaci-infested* collards for 1 week. Additional seedlings were infested with *B. tabaci* and *N. tenuis* by introducing adult mirids (1 adult/plant) along with the infested collard plants. As above, the mirids were provided with *E. kuehniella* eggs and *Artemia* spp. cysts to ensure survival and maximum oviposition. The infested seedlings were transplanted on 4 March 2016, with a 30.5 cm spacing between plants. There were two treatments: tomato alone (40 plants per plot) and sesame-tomato interplanting (10 sesame plants next to 20 tomato plants per plot; Figure 3A). Treatments were separated by 14 tomato plants sprayed on 19 April 2016 with Dinotefuran (Venom 118 mL/4047 m^2^). The treatments were arranged in RCB (n = 4) distributed in two rows separated by rows of corn. Whiteflies were monitored weekly by taking 1 leaflet from 50% of the plants in the plot. In the laboratory, all whitefly eggs and nymphs were counted using a dissecting microscope (10×). The presence and absence of *N. tenuis* was evaluated by counting the number of plants in the plot having at least 1 adult or nymph. 

#### 4.4.3. Field Comparison of *N. tenuis* and Pesticide to Control *B. tabaci* in a Tomato Boarded by Sesame

In the fall of 2016, the ability of *N. tenuis* to control *B. tabaci* was compared to insecticide-treated tomatoes in plantings bordered by sesame. Each treatment plot had 30 grape tomatoes (‘BHN784′) bordered by 10 sesame plants and was assigned to 1 of 3 treatments (n = 4): (1) untreated tomato, (2) insecticide-treated tomato or (3) tomato + *N. tenuis*. Seedlings were transplanted on 19 September 2016, after being inoculated with *B. tabaci* and *N. tenuis* for mirid treatment plots 1 week prior, as described above (Figure 4A). Initial field samples indicated that populations of *B. tabaci* were low, and additional releases were made by placing 1 *B. tabaci* infested tomato plant (var. ‘Lanai’) per plot on 24 October 2016. Mirid numbers were also low and an additional release of 1 *N. tenuis* per plant was performed on October 31 in the designated mirid treatment plots. The insecticide-treated tomato plots and the sesame bordering the untreated tomato plots were drenched with imidacloprid (Admire^®^ Pro 207 mL/ 4047 m^2^). The sesame was treated with pesticides to form a barrier to help limit the possible spread of *N. tenuis* to the untreated tomato plots. Acibenzolar-S-methyl (Actigard^®^ 10 mL/ 4047 m^2^, Syngenta Crop Protection, LLC, Greensboro, NC, USA) and extract of *Reynoutria sachalinensis* (Regalia^®^ 1 qt/ 4047 m^2^, Marrone Bio Innovations, Inc. Davis, CA, USA) were applied to all plots on 5 October, 10 October, 18 October, 25 October, 1 November, 8 November, and 15 November 2016 for suppression of the fungal and bacterial disease. A single application of *Bacillus thuringiensis* subspecies *aizawai* strain ABTS-1857 (XenTari, 1 lb/ 4047 m^2^, Valent BioSciences LLC, Libertyville, IL) was applied to all plots on Oct. 5 to control southern armyworm (*Spodoptera eridania* (Stoll), Lepidoptera; Noctuidae). *Bemisia tabaci* numbers were monitored weekly beginning Oct. 3 by selecting 3 leaflets randomly from the mid-canopy of all sesame plants per subplot and 9 leaflets per plot from tomato plants. In the laboratory, all *B. tabaci* eggs and nymphs were counted using a dissecting microscope (10×). The presence and absence of *N. tenuis* was evaluated on 17 October, 28 November, and 29 December 2016, by counting the number of plants in the plot having at least 1 adult or nymph.

#### 4.4.4. Field Evaluation of *M. praeclarus* in a Tomato-Sesame Intercrop 

This field experiment evaluated the ability of *M. praeclarus* to establish and persist on tomato and whether intercropped sesame functioned to maintain *M. praeclarus* in the field. Two weeks prior to planting in the field, grape tomato (Jolly Girl) and sesame seedlings were infested with whiteflies by placing highly infested collard leaves taken from the colony into cages with the seedlings. The following week, *M. praeclarus* adults (1 per plant) were released into the cages and allowed to oviposit. As above, the mirids were provisioned with *E. kuehniella* eggs and *Artemia* spp. cysts. The infested seedlings were transplanted on 20 March 2017 (Figure 5A). Each plot held a single row of 30 plants (46 cm spacing): 30 tomato plants (tomato alone treatment) or 25 tomato plants interplanted with 5 sesame plants (i.e., 1 sesame every 5 tomato plants, tomato-sesame intercrop treatment). The plots were separated by 14 tomato plants treated with Cyantraniliprole (200 mL/ 4047 m^2^, Verimark^®^ FMC Corporation, Philadelphia, PA, USA) to exclude insects. The experiment had 2 treatments with *B. tabaci* and *M. praeclarus*-infested plants (n = 4); (1) tomato alone and (2) tomato-sesame intercrop. Each week 30% of the plants were sampled by taking one leaf per plant (8 tomato and 2 sesame leaves) and counting all *B. tabaci* nymphs and eggs in the laboratory. On the last sample date (23 May 2017), the presence and absence of mirids were assessed by counting the number of plants with at least one adult or nymph mirid. A sample of mirids was collected from each plot and identified in the laboratory. 

#### 4.4.5. Field Comparison of Pesticide, *M. praeclarus* and *N. tenuis* to Control *B. tabaci* in Tomato Alone and Tomato-Sesame Intercrop

In the spring of 2018, a field study was conducted to study the ability of *M. praeclarus* and *N. tenuis* to control *B. tabaci* in tomato. The previous field studies showed that intercropping sesame with tomato helped retain *N. tenuis*, but the presence of sesame did not help to retain *M. praeclarus*. Therefore, tomato-sesame intercrop treatment plots were established with only *N. tenuis*. Each treatment plot held 30 plants (60 cm spacing) bordered by 10 corn plants (Figure 6A). There were 5 treatments (n = 5): (1) untreated tomato (grape tomatoes, var. Jolly Girl), (2) insecticide-treated tomato, (3) tomato alone + *N. tenuis*, (4) tomato alone + *M. praeclarus* or (5) tomato-sesame intercrop + *N. tenuis* (5 tomato plants:1 sesame plant). To establish the whiteflies in the treatment plots, tomato and sesame seedlings were placed inside 24 × 61 × 91 cm cages two weeks prior to planting in the field and infested with whiteflies by adding collard leaves taken from the colony. One week later, 1 mirid per plant was released into cages with plants designated for treatments with mirids. The seedlings were transplanted into the field on 13 March 2018. Two weeks after field planting, Verimark (90 mL/plant) was applied to the pesticide treatment plots. To suppress fungal and bacterial disease, extract of *Reynoutria sachalinensis* was applied to all plots on 27 March, 9 April, and 16 April 2018 at a rate of 0.95 L per 76, 151, and 227 L of water per 4047 m^2^ per date, respectively. The fungicide Actigard was applied to all treatments on 27 March, 9 April, and 16 April 2018 at a rate of 10 mL per 76, 151, and 227 L of water per 4047 m^2^ per date, respectively. Each week, *B. tabaci* populations were monitored by sampling 1 leaflet from the mid-canopy of 6 randomly selected plants per plot. In the laboratory, all eggs and nymphs were counted using a dissecting microscope (10×). The presence of mirids was determined weekly by counting the number of plants with at least one mirid adult or nymph per plot. At the end of the experiment, mirid feeding damage was assessed by counting the number of necrotic rings found on 5 tomato stems taken from the top one-third of the canopy of 5 randomly selected plants. 

### 4.5. Statistical Analysis 

*Bemisia tabaci*, mirid, and necrotic rings (mirid damage) data were tested for normality (Shapiro–Wilk test) and homoscedasticity (Bartlett test) using software from SAS Institute (JMP, Cary, NC, USA). The total number of egg and nymph *B. tabaci* and total number of adult and nymph mirids were transformed using log (x + 1) for count data and arcsine-square root for percentage data to meet the assumptions of analysis of variance (ANOVA). The data were analyzed using a two-factor nested random effects model with treatment and sampling week considered as fixed factors and plot as a random factor. Treatment was nested within plot. Pairwise comparisons of the fixed factor levels were made using the Tukey’s HSD post hoc tests. *p*-values ≤ 0.05 were considered statistically significant.

## Figures and Tables

**Figure 1 plants-11-02779-f001:**
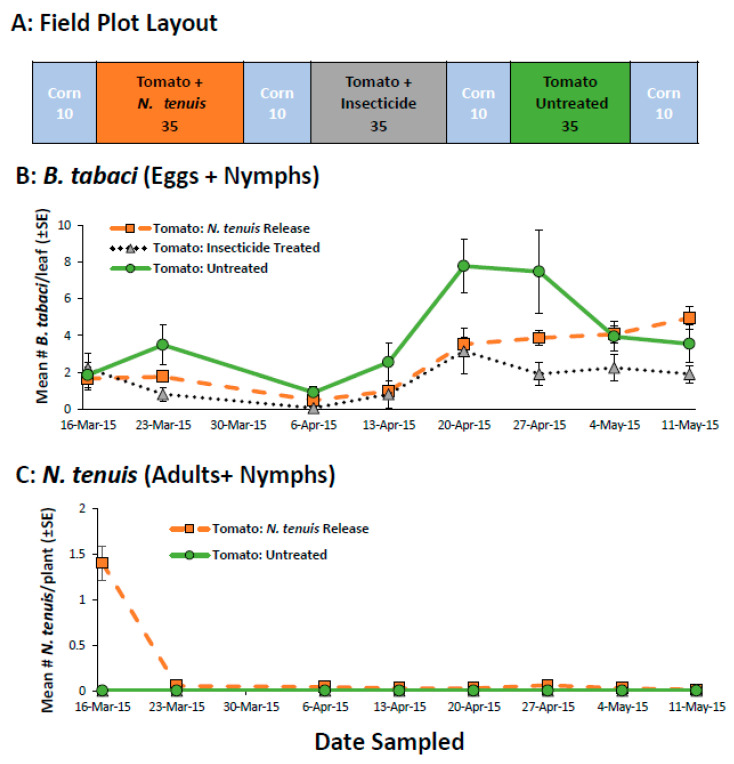
The field plot layout (**A**), the mean (±SE) number of *B. tabaci* (eggs + nymphs) per leaflet (**B**), and the mean (±SE) number of *N. tenuis* (adults + nymphs) per tomato plant (**C**) in a study that compared the ability of *N. tenuis* to control *B. tabaci* to insecticides (imidacloprid). The number under each field plot treatment (**A**) indicates the number of tomato plants.

**Figure 2 plants-11-02779-f002:**
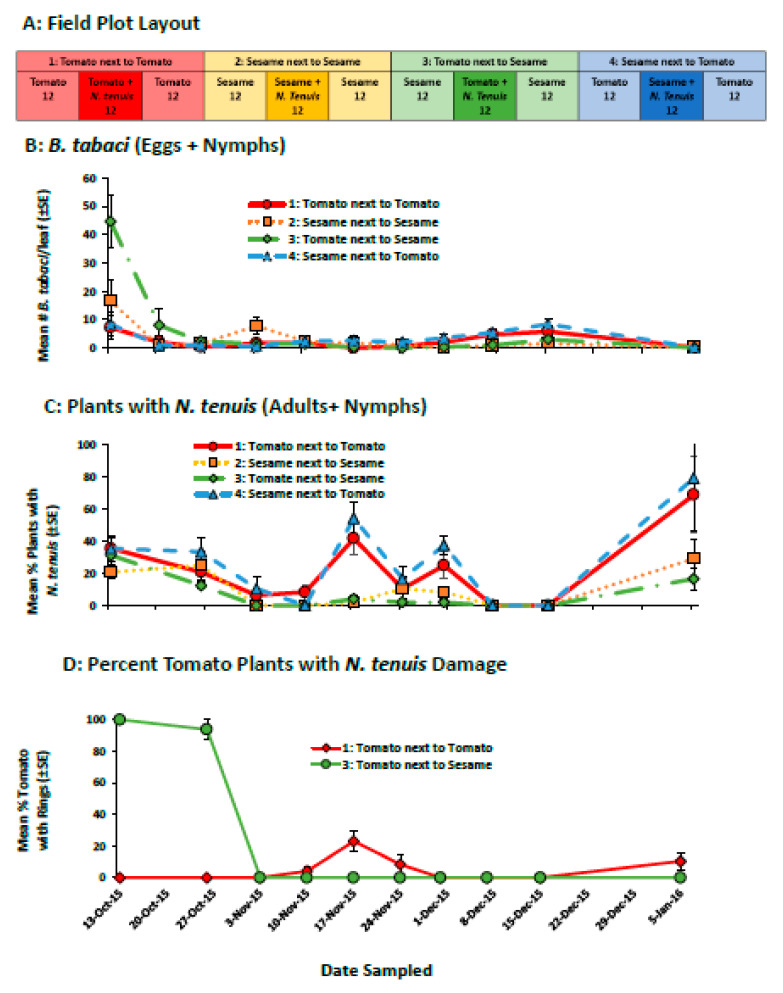
The field plot layout (**A**), the mean (±SE) number of *B. tabaci* (eggs + nymphs) per leaf (**B**), the mean (±SE) percentage of plants with *N. tenuis* (adults + nymphs) (**C**), and the mean (±SE) percentage of tomato plant with *N. tenuis* damage (necrotic rings) (**D**) in a study that compared the ability of sesame alone and inter-planted with tomato to retain *N. tenuis* and to control *B. tabaci*. The number under each field plot treatment (**A**) indicates the number of tomato and sesame plants.

**Figure 3 plants-11-02779-f003:**
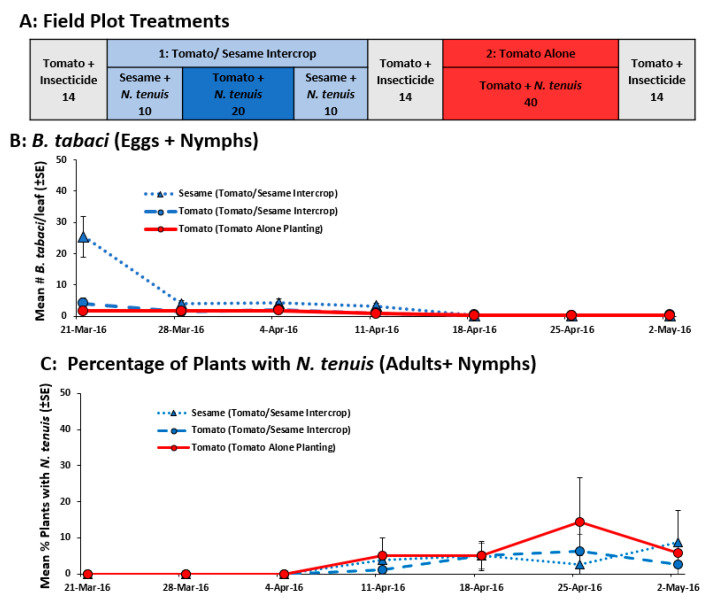
The field plot layout (**A**), the mean (±SE) number of *B. tabaci* (eggs + nymphs) per leaflet (**B**) and the mean (±SE) percentage of plants with *N. tenuis* (adults + nymphs) (**C**), in a study that compared the ability of a reduced number of sesame plants intercropped with tomato to retain *N. tenuis* and to control *B. tabaci*. The number under each field plot treatment (**A**) indicates the number of tomato and sesame plants.

**Figure 4 plants-11-02779-f004:**
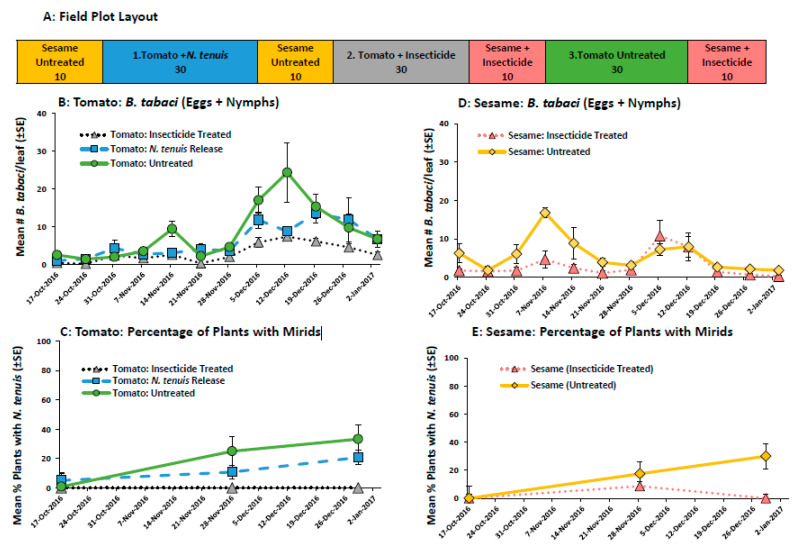
The field plot layout (**A**), the mean (±SE) number of *B. tabaci* (eggs + nymphs) per tomato (**B**) and sesame leaflet (**C**) and the mean (±SE) percentage of tomato (**D**) and sesame (**E**) plants with *N. tenuis* (adults + nymphs), in a study that compared insecticide-treated and *N. tenuis* releases to control *B. tabaci*. The number under each field plot treatment (**A**) indicates the number of tomato and sesame plants.

**Figure 5 plants-11-02779-f005:**
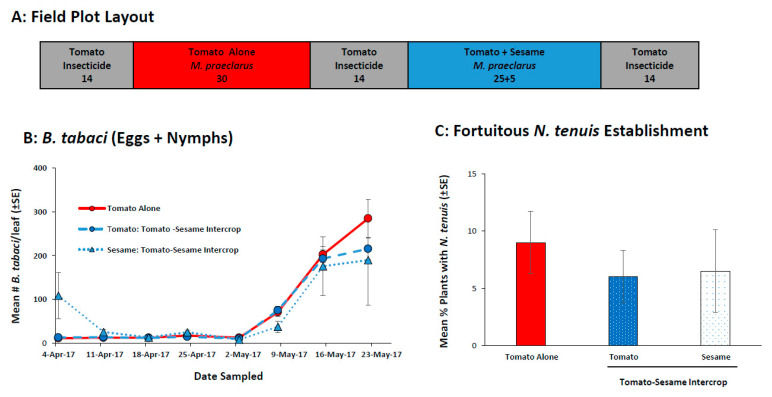
The field plot layout (**A**) and the mean (±SE) number of *B. tabaci* (eggs + nymphs) per tomato and sesame leaflet (**B**) in a study that tested whether a tomato-sesame intercrop retained *M. praeclarus*. *M. praeclarus* were originally released but not recovered and *N. tenuis* colonized from the surrounding area are reported as the mean (±SE) percentage of tomato (**C**) plants with *N. tenuis* (adults + nymphs). The number under each field plot treatment (**A**) indicates the number of tomato and sesame plants.

**Figure 6 plants-11-02779-f006:**
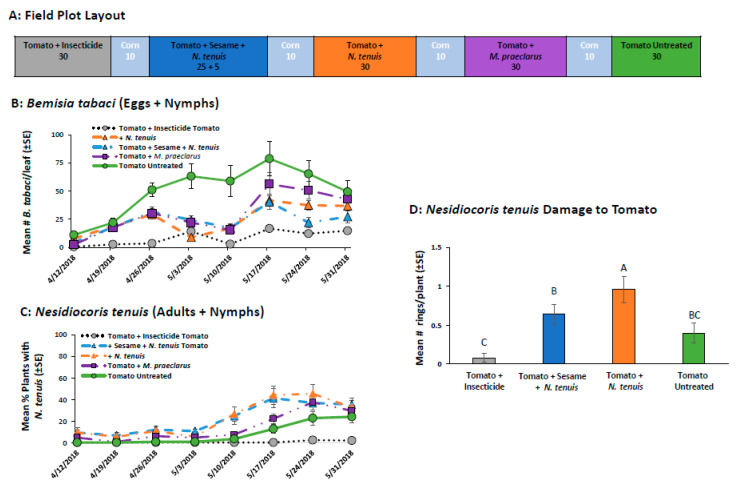
The field plot layout (**A**) and the mean (±SE) number of *B. tabaci* (eggs + nymphs) per tomato (**B**) and percentage of sesame plants with mirids (**C**) in a field study that compared insecticides, *M. praeclarus* and *N. tenuis* releases to control *B. tabaci* in tomato alone and in a tomato-sesame intercrop. *N. tenuis* invaded all mirid release plots and mirid damage measured as the mean (±SE) number of necrotic rings (**D**) was caused by *N. tenuis*. The number under each field plot treatment (**A**) indicates the number of tomato and sesame plants.

## Data Availability

All data included in the main text.

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
