# Peer review of "Sesame as an Alternative Host Plant to Establish and Retain Predatory Mirids in Open-Field Tomatoes"

_plants, 2022, doi:10.3390/plants11202779_

Round 1
Reviewer 1 Report
Dear corresponding authors,
your manuscript "Sesame as an alternative host plant to establish and retain predatory mirids in open-field tomatoes" submitted for publication in "Plants" presents very interesting data from a detailed, well described, and analyzed study. I have no suggestions regarding these parts of your manuscript.
Starting to read your manuscript with Tuta aboluta in the very first sentence , however, I expected more on T. absoluta control by the tested mirids than on Bemisia. Your study in its current form is nevertheless a detailed report on the efficiency of both mirids as control agents against B. tabacci. This is definitely important and worth publishing, but it should be mentioned, introduced, and discussed as such.
Please explain why you did not conduct research on T. absoluta eventhough the initial focus in Abstract and Introduction was put on this invasive pest species.
Moreover, in the key words Bemisia is entirely missing.
The overlap of your study with the published study by Konan et al. 2021, that you cited, is very strong. So please clearly define the differences among the two studies and underline the objectives of your study in comparison with theirs.
I consider your manuscript could be re-submitted after a thorough revision regarding the focus and objectives of your study.
Author Response
The authors appreciate the comments of Reviewer 1 and the manuscript has been improved by addressing the reviewers concerns about the focus of the introduction and discussion on T. absoluta rather than B. tabaci.
Starting to read your manuscript with Tuta aboluta in the very first sentence, however, I expected more on T. absoluta control by the tested mirids than on Bemisia. Your study in its current form is nevertheless a detailed report on the efficiency of both mirids as control agents against B. tabacci. This is definitely important and worth publishing, but it should be mentioned, introduced, and discussed as such.
- Please explain why you did not conduct research on T. absoluta even though the initial focus in Abstract and Introduction was put on this invasive pest species.
We did not conduct research on T. absoluta because the pest, fortunately, has not reached the U.S.A. We are very worried about T. absoluta, particularly as the pest is moving through Central America and the Caribbean. Perhaps this fear biased our writing and influenced our focus on T. absoluta rather than B. tabaci. We have updated the abstract, introduction and discussion to make it more apparent that our focus was twofold: 1) to develop a biological control option for an existing pest, B. tabaci as well as 2) prepare for the imminent threat of T. absoluta.
- Moreover, in the key words Bemisia is entirely missing.
Bemisia was added to the key words
- The overlap of your study with the published study by Konan et al. 2021, that you cited, is very strong. So please clearly define the differences among the two studies and underline the objectives of your study in comparison with theirs.
The discussion was updated to provide more details how the Konan et al. 2021 studied differed from the studies described in this manuscript. The largest difference lies in that we tested the use of sesame in an entirely open system that better represented commercial production in the U.S.A. Konan et al. 2021 studies were conducted in cages where the mirids were forced to remain and there was no natural invasion of the pest. Our studies indicate that the benefits of sesame shown in the Konan et. al 2021 study maybe applicable to open-field production. Our study had a more practical focus in that we simply wanted to develop a method to help retain the mirids throughout the entire season and control the established pest, B. tabaci. It is well established N. tenuis is a good biological control agent for T. absoluta so we focused our efforts on this species to meet our second goal of devising alternative management strategies in anticipation of a T. absoluta invasion. Our manuscript covers the often messy aspects of an open system such as the unpredictability of the target pest (i.e. in our studies natural B. tabaci infestations were very low) and that the mirids when not confined will move to tomato plants with prey (including our untreated controls and insecticide treated plots). Our study was conducted over multiple years and in multiple seasons and the general patterns remained consistent (sesame retained the predator, there was little damage caused by N. tenuis, and N. tenuis moved from the sesame to tomato and reduced B. tabaci numbers), which will help guide the needed future studies to further test the benefit of mirid predators in controlling the established pest. We also hope that this work will help prepare growers and provide options should T. absoluta spread to new areas. We also report that sesame does not provide the benefits to M. praeclarus as seen with N. tenuis and more work is needed to determine whether this species may work in open field tomatoes. Finding N. tenuis in the M. praeclarus plots opens the idea for the possible negative interactions the species may be having particularly when prey is naturally low. Again, future studies are needed to address these questions. This work provides a starting point for developing strategies to use mirid as biological control agents in large open-field production systems.
Reviewer 2 Report
The paper deals with an important topic related to the BC of tomato pests and offers interesting indications for the management strategy of established and potentially invasive species on this crop.
Minor notes are reported in the reviewed MS
I would just pinpoint the following considerations:
I am not sure the layout of the experimental trial presented in Fig 4 is correct; if yes it should be explained why is asymmetric and conceived this way (tomato untreated sided with treated sesame)
some previous results (highlighted in the reviewed MS) don't seem in strictly accordance with these results: This should be better discussed
Different tomato varieties were used in the different trials. This choice should be justified
In the complex the MS fully deserves publication

Author Response
We very much appreciate the time and effort that Reviewer 2 took to make a careful and detailed review. The manuscript has improved greatly from the comments.
- Minor notes are reported in the reviewed MS
All minor corrections were made to the manuscript as suggested by the reviewer.
- I am not sure the layout of the experimental trial presented in Fig 4 is correct; if yes it should be explained why is asymmetric and conceived this way (tomato untreated sided with treated sesame)
The layout in Fig. 4 is correct despite it looking strange and asymmetric. The sesame next to the untreated tomato were treated with pesticides as a means to prevent (or at least slow) the movement of the mirids from the release plots into the untreated tomato. We wanted to compare the number of whiteflies found on untreated tomatoes to mirid release plots but found it difficult to exclude the mirid predator from moving to these plots that had more prey. The methods section was updated to include this explanation in the field design.
- some previous results (highlighted in the reviewed MS) don't seem in strictly accordance with these results: This should be better discussed
We believe that M. praeclarus still may be a good candidate for biological control in open field tomatoes. We are not sure why we did not recover M. praeclarus but suspect it may have to do with the availability of prey and that sesame really wasn’t a good alternative plant for this species. There is always the possibility that N. tenuis displaced or preyed upon M. praeclarus. All these factors could have had an impact on M. praeclarus populations. We have discussed these issues further in the manuscript.
We also are currently running more experiments with M. praeclarus, further north in Florida and in an area where N. tenuis has yet to be reported. We also are trying different companion plants. Hopefully these results will help us understand whether or not M. praeclarus is a good candidate for biological control in open field tomatoes. Hopefully good results to share soon.
- Different tomato varieties were used in the different trials. This choice should be justified
Different varieties were used to better match the seasonal conditions of the particular Florida growing season (i.e. spring vs fall). The varieties selected were ones commonly being used by the growers and readily available and selected to withstand disease and pest problems outside this study (i.e. viruses and nematodes). The methods have been updated with this reasoning as well as reference provided that lists the seasonal selections.
Reviewer 3 Report
The MS "Sesame as an alternative host plant to establish and retain predatory mirids in open-field tomatoes" is a well written manuscript. Materials and methods, results and discussion sections are well written and are adequate. . Introduction is little long and may be shortened. I do not see any concerns with the MS.
Author Response
The authors appreciate the effort made by Reviewer 3.
- The MS "Sesame as an alternative host plant to establish and retain predatory mirids in open-field tomatoes" is a well written manuscript. Materials and methods, results and discussion sections are well written and are adequate. Introduction is little long and may be shortened. I do not see any concerns with the MS.
We have improved the manuscript by shortening the introduction through removing details that a reader could easily find in the references provided.
Reviewer 4 Report
Sesame as an alternative host plant to establish and retain predatory mirids in open-field tomatoes
The manuscript deals with a field investigation carried out over four years, with the aim of developing methods to display and maintain zoophytophagous mirid insects in open field tomatoes, in preparation for an eventual invasion of Tuta absoluta in free areas of this pest, both in the United States as in Mexico. As a secondary objective, the authors also evaluated the efficacy of the proposed technological alternative (the myriid species and the use of sesame, Sesamum indicum L) in the control of Bemicia tabaci. Although the final scope of the MS is to have a sustainable alternative to control an eventual invasion of T. absoluta to the tomato production area in USA, the objective carried on is more focused on how to retain two species of zoophytophagous/predatory mirids to control B. tabaci.
The experimental work looks carefully planned and very well executed; statistical analysis are adequate, and the results obtained are valuable because data of this type are scarce and not easy to acquire. I just recommend two items: 1) as B. tabaci is the main pest of the tomato in Fla, authors should discuss deeply how good was the control of the tested mirids on B. tabaci, compared to current used strategies. This acquires special importance in relation to the work done, because although T. absoluta appears as the central key species of this research, it was absent in the experimental work. 2) A second question could be How difficult would it be to maintain myrid density in the open field to achieve levels of control similar to insecticide treatments? The authors should discuss further what can be expected, under large open field conditions, from the use of mirids plus the inclusion of S. inducum in terms of control of B. tabaci and, of course, an eventual invasion of T. absoluta.
Below I listed some minimal observations:
Line 151, change poplutions by populations ..
Figure 2, the legends of the field plot layout are not easy to read. Not legible even with high augmentation
Legend of figure 2, change leaflet by leaf
Lines 297-300, this is a "double-edged knife", since being generalists they may not attack the most important pest and therefore not achieve good control. This must also be discussed.
Author Response
The authors very much appreciate the time and effort Reviewer 4 made and the manuscript has been improved from the thoughtful comments.
1) as B. tabaci is the main pest of the tomato in Fla, authors should discuss deeply how good was the control of the tested mirids on B. tabaci, compared to current used strategies. This acquires special importance in relation to the work done, because although T. absoluta appears as the central key species of this research, it was absent in the experimental work.
In all 4 years, N. tenuis kept white fly numbers very low and where similar when compared to insecticide treatments. What was interesting, and challenging, was that whitefly numbers were consistently very low in all treatments and the mirids moved into the untreated plots and insecticide treated plots towards the end of the experiment. This made any final conclusions on the impact of the mirid to the pesticides difficult to compare so we focused on describing what could justifiably be said and supported by the data.
Based on Reviewer 4 concerns, the discussion has been updated to better describe how our experiments showed that N. tenuis controlled B. tabaci numbers to similar levels found in the plots where standard pesticide applications were made.
Additionally, the introduction was updated to better describe our goals of the study, 1) to develop a biological control option for an existing pest, B. tabaci and 2) to prepare for the potential incursion of T. absoluta. We are very worried in the U.S. about T. absoluta and are working hard to prepare. We were fortunate to have N. tenuis, a known good bc agent of T. absoluta, already established in the U.S. Our fear of T. absoluta likely biased our focus to this pest. We appreciate Reviewer 4 reminder of the importance of the work on B. tabaci biological control.
2) A second question could be How difficult would it be to maintain myrid density in the open field to achieve levels of control similar to insecticide treatments? The authors should discuss further what can be expected, under large open field conditions, from the use of mirids plus the inclusion of S. inducum in terms of control of B. tabaci and, of course, an eventual invasion of T. absoluta.
We have included further discussion on what could be expected under large open field conditions. The challenge will be in integrating a companion crop into the existing cultivation and very pesticide reliant system. We tested reduced levels of sesame as we understand that growers must use as much space as possible for the crop. Our results are hopeful and indicate that not many plants are needed. We also saw that N. tenuis moved into a M. praeclarus experiment from the surrounding area, indicating that having a refuge for them by providing sesame may work. This is just a small part of what needs to be known to begin answering the question posed by the reviewer. There is much more work needed to determine the spatial distribution of the sesame in open field systems, perhaps along irrigation lines or at the boarder of the field will be sufficient. We plan to continue studies that are more representative of a large field now that we have learned better what may work to retain them in the area.
3) Below I listed some minimal observations:
Line 151, change poplutions by populations ..
Figure 2, the legends of the field plot layout are not easy to read. Not legible even with high augmentation
Legend of figure 2, change leaflet by leaf
All minor changes were made as suggested by Review 4
4) Lines 297-300, this is a "double-edged knife", since being generalists they may not attack the most important pest and therefore not achieve good control. This must also be discussed.
We have included further discussion on the important possibility of a generalist predator not attacking the target pest in the manuscript. We pointed out that even though the predators we tested are generalists that have show a preference to prey, particularly to our target pests (N. tenuis for T. absoluta and Macrolophus spp. to whiteflies). We have included the references to these studies and pointed out the importance of understanding the generalist